# Cystic Echinococcosis in the Early 2020s: A Review

**DOI:** 10.3390/tropicalmed9020036

**Published:** 2024-01-31

**Authors:** Mihai-Octav Hogea, Bogdan-Florin Ciomaga, Mădălina-Maria Muntean, Andrei-Alexandru Muntean, Mircea Ioan Popa, Gabriela Loredana Popa

**Affiliations:** 1Department of Microbiology II, Carol Davila University of Medicine and Pharmacy, 020021 Bucharest, Romania; mihai-octav.hogea@drd.umfcd.ro (M.-O.H.); bogdan-florin.ciomaga@drd.umfcd.ro (B.-F.C.); madalina.muntean@umfcd.ro (M.-M.M.); alexandru.muntean@umfcd.ro (A.-A.M.); mircea.ioan.popa@umfcd.ro (M.I.P.); 2Department of Microbiology, Cantacuzino National Military Medical Institute for Research and Development, 050096 Bucharest, Romania; 3Parasitic Disease Department, Colentina Clinical Hospital, 020125 Bucharest, Romania; 4Department of Microbiology, Faculty of Dentistry, Carol Davila University of Medicine and Pharmacy, 020021 Bucharest, Romania

**Keywords:** *Echinococcus granulosus*, foodborne disease, One Health, zoonosis

## Abstract

Cystic echinococcosis (CE) is a zoonosis caused by metacestodes, the larval stage of *Echinococcus granulosus*. Although the World Health Organization (WHO) has defined CE as a neglected disease, it is the second most important foodborne parasitic disease, and it remains an important public health issue, considering its zonal endemicity and potential morbidity. The control and prevention of CE is a relevant WHO target, especially from a One Health perspective, as the disease affects not only animals and humans but also the food chain. Since not all countries have a CE surveillance strategy or reporting system and specific management guidelines, recent epidemiological data are relatively scarce, and research concerning the specific geographical distribution of the disease is ongoing. To add new information to the subject, we have analyzed and collected data from national guidelines and several medical databases. Out of the 751 research articles that were originally identified, only 52 were included in the investigation after applying specific inclusion and exclusion criteria. Notable international projects that have provided significant contributions and had a positive impact are presented. The available data were correlated with WHO recommendations on the subject, thus showcasing the measures taken and those that are still needed to properly control the disease’s spread.

## 1. Introduction

*Echinococcus granulosus* is a zoonotic parasite that is responsible for human cystic echinococcosis (CE) [1,2,3]. The infection occurs when hosts ingest *Echinococcus* eggs, which then develop into the larval (metacestode) stage. Regarding the parasite’s host, canids are the definitive host of the adult-stage parasite [4,5]. The parasite’s metacestode (hydatid) stage thrives in domestic ruminants, such as sheep, cattle, and camels [3]. Transmission from definitive to intermediate hosts occurs via the fecal–oral route. Because humans do not biologically support the parasite’s life cycle, they represent accidental dead-end intermediate hosts [6].

Cystic echinococcosis is diagnosed primarily based on imaging, clinical presentation, and serology. The gold standard for imaging in the diagnosis of abdominal CE is ultrasonography, while computer tomography, magnetic resonance imaging, and conventional radiography can also prove useful in specific circumstances. In this manner, cysts can be classified into categories based on their appearance. To achieve global disease assessment consistency, the World Health Organization Informal Working Group on Echinococcosis (WHO-IWGE) has classified CE cysts into five types and three groups, with CE1 and CE2 reflecting active infection, CE3 being a transitional stage, and CE4 and CE5 representing inactive cysts [1]. Imaging is also used to monitor these patients.

In terms of clinical appearance, the gradual development of the cysts renders patients asymptomatic for an extended period, delaying the diagnosis. Symptoms in humans vary depending on the cysts’ location and size, as well as their number [2,4,7]. Throughout the disease, the liver is the most common location for cysts [4]. Most patients have only one organ affected with a solitary cyst, but in some unfortunate and severe instances, multiple cysts can develop in various viscera [8,9,10,11,12,13], such as the lungs, kidneys, spleen, brain, and bones.

Major antigenic compounds of the hydatid fluid, such as antigen B and antigen 5, can be immuno-assessed. Serum antibodies can be detected using enzyme-linked immunosorbent assay (ELISA), indirect hemagglutination (IHA), and latex agglutination with antigens from the hydatid cyst fluid, with varying sensitivity. ELISA and immunochromatography performed better than IHA as a complementary to the imaging diagnosis [14]. Immunoblotting is usually used in differential diagnosis. As a rule, it is recommended that serology and ultrasonography imaging be used together when performing mass screening [1,15].

The genus *Echinococcus* (Cestoda: Taeniidae) has undergone multiple taxonomic revisions since the 1960s [16]. In the past, *Echinococcus granulosus* included up to nine sub-specific genotypes (G1–G9) [2,16,17] or strains that evolve into CE during the metacestode stage. It is currently proposed that the *Echinococcus* genus be classified into at least nine different species [2,18]. All *Echinococcus* species that are capable of causing CE in intermediate hosts can be referred to collectively as *E. granulosus* sensu lato (s.l.), while strains G1–G3, which are closely related, can now be categorized as *E. granulosus* sensu stricto (s.s.) [16,18,19].

Although echinococcosis has historically been described as endemic in Europe, the World Health Organization (WHO) considers the infection to be one of the 20 neglected tropical diseases of great public health significance [20]. A comprehensive understanding of the infection’s pathogenesis and its current epidemiological landscape is needed to prevent its further spread. To accomplish this, the WHO published a document entitled “Ending the neglect to attain the Sustainable Development Goals: A road map for neglected tropical diseases 2021–2030” [20], which presents worldwide statistics on echinococcosis and other neglected tropical diseases (NTDs). In light of this endeavor, the WHO aims to reach global disease control by 2030. Specialists worldwide have taken notice of the hyperendemic areas, and an increasing number of countries have joined the WHO efforts to control it [21,22,23].

This review contributes to the discussion on cystic echinococcosis by zooming in on recent findings, particularly in epidemiology, and by correlating available data with WHO recommendations on the subject, thus showcasing the measures that are taken and those that are still needed to properly control the disease’s spread.

## 2. Materials and Methods

The data included in the paper were revised in November 2023. Within the scope of our review, we have analyzed and collected data on the subject from national guidelines, as well as several well-established medical databases—Web of Science, PubMed, Cochrane, ScienceDirect, PubMed Central (PMC), and Google Scholar.

National guidelines included documents published by a nationally recognized committee such as French Institute for Public Health Surveillance (INVS), German Center for Infection Research (DZIF), Italian National Centre for Disease Prevention and Control (CCM), Romanian National Center for Surveillance and Control of Infectious Diseases (CNSCBT), United Kingdom Health Security Agency (UKHSA), or the US Centers for Disease Control and Prevention (CDC).

The Medical Subject Headings (MeSH) technique was used, and parallel strategies employing identical keywords were used in the other available databases (Web of Science, Cochrane, ScienceDirect, PubMed Central, and Google Scholar). The papers were first included based on titles and abstracts, followed by full-text analysis of relevant articles. We used keywords such as “*Echinococcus granulosus*”, “*E. granulosus*”, “echinococcosis”, “cystic echinococcosis”, “prevalence”, “incidence”, “diagnosis”, “clinical presentation”, “treatment”, and “prevention”, both on their own and with Boolean operators to narrow down the search. Other common terms associated with CE (i.e., “hydatid disease”, “hydatid cyst”, etc.) were also included in the database search, but did not yield any additional findings.

The inclusion criteria for our research included manuscripts that offered relevant information regarding cystic echinococcosis, such as taxonomic changes, epidemiological data (i.e., human incidence rates (IRs) and prevalence rates (PRs)), treatment options, and new research contributing to the control and prevention of the disease (i.e., new biomarkers used for diagnosis, etc.).

Exclusion criteria included incorrect agents (e.g., *Echinococcus multilocularis*), studies not pertaining to human hosts, studies not written in English, review articles devoid of original data, editorials or letters to the editor that were devoid of original data, and/or articles lacking an IR or PR estimate for the epidemiology search.

## 3. Results

The total number of research articles that were obtained through database searches of the titles and abstracts was 751. After eliminating duplicates (*n* = 53), we further narrowed down our database by ruling out papers that described *Echinococcus multilocularis* (*n* = 194), as well as those studies not pertaining to human hosts (*n* = 161) and those not written in English (*n* = 29).

The remaining manuscripts (*n* = 314) were then reviewed in their entirety, at which point, articles (*n* = 121), editorials, and letters to the editor (*n* = 93) that were devoid of original data, as well as studies lacking IR or PR estimates (*n* = 11), were further removed.

In the course of our investigation, we consulted additional guidelines on the topic of CE management. For more details, please refer to the Appendix A.

In total, 89 studies were included in our investigation. Figure 1 shows the selection process. Additional information pertaining to the included studies is available in Appendix A.

### 3.1. Taxonomy

The genus of pathogens that we now define as *Echinococcus* has undergone several changes in the past, owing mostly to its transfer to humans from multiple different hosts and the variable morphology of the metacestodes. After its formal classification as *Echinococcus*, there was some debate as to whether or not the clinical infection with *E. granulosus* and *E. multilocularis* was caused by the same pathogen. This issue was resolved when a distinct account of the life cycle of *E. multilocularis* was provided. Afterwards, the *Echinococcus* genus was split into four species, with *E. oligarthra* and *E. vogeli* joining the two previously listed [18].

*E. granulosus* seemed to contain a high number of variations, hence its designation as *E. granulosus* sensu lato. Through studies of molecular epidemiology and geographical data over the course of two decades, it became clear that the genotypes that were enclosed within *E. granulosus* sensu lato were significantly diverse, and that the genotype taxonomy model was becoming increasingly limited and contradictory [18,24]. The development of mitochondrial DNA sequencing led to the description of 10 genotypes (G1 to G10), a classification that was heavily debated [18,24]. The term *E. granulosus* sensu lato now includes the *E. granulosus* sensu stricto genotypes (G1–G3), as well as *E. equinus* (G4), *E. ortleppi* (G5), *E. intermedius* (G6, G7), *E. canadensis* (G8, G10), and *E. felidis* [18,24,25,26,27].

### 3.2. Epidemiology

Transmission and Life Cycle.

In terms of transmission, echinococcosis depends on the presence of different hosts within endemic regions, both definitive (domestic dog, lion, etc.) and intermediate (cattle, pig, sheep, etc.). A series of transmission routes have been described, ranging from the fecal–oral route—consumption of contaminated water, unwashed raw produce, and contact with contaminated soil—as well as contact with dogs and livestock (particularly ruminants), either through direct contact or through contact with contaminated fur [6,28,29].

When discussing echinococcosis, it is particularly important to consider that the transmission method varies geographically, depending on host availability, social and cultural customs, awareness of public health, and environmental conditions. Although direct contact with canine hosts has long been suspected as the main source of transmission to humans, the correlation is not compelling. There are endemic regions where the disease’s prevalence and the population of infected dogs are unrelated, or where contact with dogs in the area is kept at a minimum despite their presence. This has led to the conclusion that, while direct contact is not always the cause, the high environmental quantity of *Echinococcus* eggs contributes to transmission, likely from soil contamination [6,30,31].

After transmission, the eggs enter their larval stage, developing into metacestodes. The metacestodes then grow within the intermediate hosts’ organs, with a preference for the liver and lungs. The ingestion of these infected organs by canines leads to the development of adult worms, thereby completing the cycle [6,28]. Because of the multiple paths of transmission, along with the inability to accurately determine the time of infection based on the metacestode itself, determining the precise moment and method of transmission in a patient is difficult [6,29].

Epidemiological Data.

Cystic echinococcosis is regionally endemic throughout Europe, North and East Africa, Central Asia, the Middle East, Central and South America, and Australia, particularly in areas with significant animal husbandry and livestock farming [2,28,32]. The WHO estimates an incidence of more than 50 per 100,000 person-years, a prevalence of 5–10% in certain endemic regions, a 20–95% prevalence of CE in slaughtered livestock, as well as an estimated 1 million people currently suffering from this zoonosis [2]. 

Despite being classified as a neglected tropical disease, studies regarding CE distribution, as well as the interventions that are aimed at disease prevention and control, have been ongoing for some time. As an example, consider Kenya, where the country’s population living in the Turkana region has the highest disease prevalence globally (between 5 and 10 percent) after screening protocols based on ultrasound and serodiagnosis were implemented in 1986 [33]. In 1989, a study was undertaken in the East African region (Kenya, Sudan, Ethiopia, and Tanzania) to determine the prevalence of the disease. The study involved 18,565 participants and utilized an ultrasound screening methodology. In the investigated population, the average prevalence rate was 2.2%, whereas individuals with risk factors had a rate of 3.2% [34]. According to a 2017 study, disease prevalence rates have been consistently declining, with some regions, such as Turkana, Kenya, experiencing rates as low as 1.9% between 2010 and 2012 compared to an initial prevalence of around 10%, highlighting the efficacy of the employed control and prevention strategies [35].

The latest study regarding the global distribution of cystic echinococcosis was published in 2017 [36]. The study not only encompassed a meticulous analysis of the disease’s prevalence in both animal and human populations but also discussed the incidence rates within the human population. Remarkably, the collected data revealed a significant time gap in the examination of cystic echinococcosis in certain animal hosts, given that the most recent studies for some of these hosts trace back to 1956 [36,37]. This significant gap in investigations underlines a fundamental aspect contributing to the neglected status of cystic echinococcosis. Furthermore, the disparity in incidence between regions is noteworthy, with non-endemic regions having gaps of even decades between confirmed cases, and with areas having a highly variable incidence. The highest mentioned incidence is over 30 per 100,000 individuals in Kyrgyzstan and Tibet. The summarized data of the study can be found in Appendix A.

A 2020 systematic review of the epidemiological distribution of *E. granulosus* s.l. revealed a mean annual incidence of up to 7.74 per 100,000, as well as a prevalence of CE of up to 64.09% in cattle, 68.73% in sheep, and 31.86% in dogs in certain areas of the Mediterranean and Balkan Europe [28]. Prior studies from the early 2000s describe a similar annual incidence of *E. granulosus*, which may indicate that the current annual incidence has reached a plateau in the areas referenced prior; both assessments, however, have improved since the 1980s, when the annual incidence was 19 per 100,000 [24,38].

The first ultrasound-focused survey that focused on determining the prevalence rate of CE in Chile was published in 2022 [39]. The research identified the prevalence rate within the studied population as 1.6% (CI 95%, 1.1–2.2), while also concluding that the risk factors associated with the infection were rural areas, old age, and the drinking of non-potable water. A total of 84.6% of the infected participants had been diagnosed during the study.

In 2023 [40], a research paper summarized the latest cross-sectional population studies, which included ultrasonography population surveys, by analyzing the cystic echinococcosis prevalence rates. The prevalence rates were divided by continent, with a focus on the prevalence of untreated CE in population-based ultrasound studies. Additional information can be found in Table 1 [40].

In terms of the socioeconomic burden, echinococcosis has been estimated to be responsible for a total annual cost of USD 3 billion, comprising both treatment and livestock costs [2].

### 3.3. Current Guidelines

#### 3.3.1. Prevention and Control

In terms of prevention and control, the CDC and WHO recommendations center around preventing exposure and actively treating potential animal hosts that are at risk of coming into contact with infectious material. This can be done by preventing canid access to infected organs and carcasses, controlling the stray dog population, discouraging home slaughter of sheep and cattle, improving sanitation concerning animal slaughter, deworming dogs with praziquantel, and implementing public health programs for rural populations that are most susceptible to this disease. A vaccine for sheep with an *E. granulosus* recombinant antigen is already in use in Argentina and China [41,42]. The WHO also offers an optimistic estimate that a proper combination of these preventative measures may eliminate CE in as little as 10 years [2,32].

Despite this, data remain scarce, and these estimations may be affected by underreporting. The systematic review mentioned previously offers a different perspective, as the article’s findings suggest that while all CE cases that end in hospitalization and some CE patients who display symptoms are reported, there may be many more instances of echinococcosis patients that are not. This is further compounded by the fact that many countries do not report their findings at a higher institutional level, which means that the overall effect of this disease on human and livestock populations, as well as the overall efficiency of the proposed preventative measures, is likely underestimated [2,28,32].

One such example can be observed in the case of Nigeria. The last study that focused on the prevalence rates of CE was conducted in 1987 and identified a seroprevalence of 0.53% within 176 hospitalized patients [43]. A recent study highlights the limitations of the current approach regarding CE, especially from an epidemiological point of view, when discussing the spread of the disease in Nigeria. The study also provides suggestions on how to address this, underscoring the importance of focusing on prevention and control measures [44].

When comparing the geographical distribution of both human and animal hosts within Europe, it was noted that the human distribution more closely followed the livestock distribution compared to the canine distribution. Living in rural areas and dog ownership were found to be the most significant risk factors, while contact with dogs had variable degrees of association. Evidence seems to suggest that environmental contamination, as well as hand-to-mouth contamination and actions that perpetuate the *Echinococcus* life cycle, like feeding dogs who are contaminated viscera, are the most impactful [6,28].

In terms of food safety, although raw consumption of vegetables and other produce can be a possible transmission method, there is no formal surveillance or a standardized method of analyzing the produce for *Echinococcus* eggs. Furthermore, such findings may be difficult to correlate with an estimated risk of infection, as the correlation between finding *Echinococcus* eggs or DNA and the former’s viability and infectivity is not yet understood [6].

Drinking water from commercial sources is linked with a lower likelihood of infection, as are open bodies of water; water-based transmission of *Echinococcus* eggs is most likely in water sources that are shared between humans and animals, especially in regions where access to water is scarce [6,45].

#### 3.3.2. Treatment

Because of its risk of recurrence, the WHO-IWGE recommends that echinococcosis be managed by using a multidisciplinary approach that is similar to that of cancer. The therapeutic and/or prophylactic treatment plan is individualized, taking into account the patient’s clinical characteristics and the experience of the medical and surgical team [1,3,4,10,12,13,27,46,47,48,49,50,51,52,53,54,55]. There are multiple treatment options, and sometimes, a combination of approaches is necessary depending on the location and size of the cysts as well as the presence or absence of complications. Long-term follow-up is also recommended [1].

Surgery:Surgery was the preferred course of treatment in the past, as it may be curative by the complete removal of the cyst (total cystectomy) [3,32]. Other surgical approaches are sub-total cystectomy and hepatectomy. Total cystectomy avoids cyst opening and therefore prevents recurrence and is the preferred option if possible. Surgery is still used for particular scenarios such as liver cysts that are secondarily infected or cysts that are located in critical areas like the brain, lungs, or kidneys. It is also the elective choice of treatment for large liver cysts, particularly those over 7.5 cm, which are likely to have biliary communication [3]. It is recommended to associate albendazole to prevent relapses.Chemotherapy:Over 2000 documented cases [1,3,4,8,12,46,56,57] have been treated with benzimidazoles. The optimal course of treatment includes albendazole (10–15 mg/kg/day or 400 mg q12h) taken after a fatty meal, alone or in association with mebendazole (40–50 mg/kg/day divided into three doses during fat-rich meals) or praziquantel (40 mg/kg once a week), with variable treatment outcomes [1,4,27,56]. Chemotherapy results in cyst disappearance (free from disease) in 10–30% of patients, improvement in 50–70%, and no change in 20–30%. It is generally more effective in younger patients and against specific cyst types. Chemotherapy is indicated in inoperable patients with primary liver echinococcosis, patients with multiple cysts in multiple organs, and in secondary echinococcosis prevention. However, chemotherapy is not recommended for large cysts at risk of rupture, inactive or calcified cysts, compromised patients with severe chronic hepatic diseases, or in early pregnancy. It can be administered before surgery for the safe manipulation of cysts, since it inactivates protoscolices, alters the integrity of cystic membranes, and reduces cyst turgidity [3,32].Chemotherapy can be given before surgery, and many regimens have been tested employing both monotherapy and drug associations. These range from treatment with albendazole administrated 1 week before surgery (continued for 2 months after surgery) and 10 mg/kg/day albendazole and 25 mg/kg/day of praziquantel for 1 month prior to surgery to facilitate the safe manipulation of cysts. It inactivates protoscolices, alters the integrity of cyst membranes, and reduces cyst turgidity [3,4,27].Puncture, Aspiration, Injection, and Re-aspiration (PAIR):PAIR [11,46,58] is a minimally invasive, ultrasound-guided cyst puncture, followed by the aspiration of cyst fluid, injection of a protoscolicidal substance (preferably 95% ethanol), and re-aspiration of the fluid after a specified time. It is used as a last resort for treating inoperable patients, relapsing instances after surgery, or non-responders to chemotherapy. A modified procedure of PAIR may also be useful as an alternative to surgery for non-complicated CE2 and CE3b cysts. The “modified catheterization technique” (MoCAT) uses sonographic and fluoroscopic guidance to aspirate both the cyst content and the parasitic membranes and to place a catheter for a period of time after the intervention [59]. Even if this method is suitable for a variety of cysts, it should not be used for lung cysts [1,3].“Watch-and-wait” Approach:The observation that some CE cysts may spontaneously become inactive leads to the withholding of treatment, as these cysts remain stable over time. The use of albendazole or other treatment options in asymptomatic patients is not recommended on a standard basis [40,47]. However, it may be an option in highly selected cases referred to specialized centers. In 2010, the WHO-IWGE published their recommendation of this approach for uncomplicated, asymptomatic, inactive (CE4-CE5 stages of the WHO-IWGE classification) CE cysts [40,47].

### 3.4. Projects and Initiatives Related to the WHO Road Map

Within the landscape of disease control and prevention, specific projects sometimes emerge as vanguards, offering innovative solutions and having a positive impact within their respective regions. Among the criteria-filtered research, we identified notable projects that have demonstrated efficacy, particularly within the European framework. Below, we will discuss the significant contributions that are made by some of these programs, highlighting their innovative approaches in investigating, controlling, and preventing CE infection.

#### 3.4.1. The HERACLES Project

The “Human Echinococcosis ReseArch in CentraL and Eastern Societies”, commonly known as the HERACLES project, was carried out between 2013 and 2018 [60,61]. The research concentrated on areas in Europe (Spain, Italy, Bulgaria, Romania) and associated countries (Turkey), where the endemic status was either suspected or confirmed by an IR of 1–200 per 100,000 inhabitants [61]. These nations enrolled patients in the European Register of Cystic Echinococcosis (ERCE), supplied parasitic and human samples to the Echino-Biobank, and collaborated on research endeavors pertaining to the molecular epidemiology of [62]. Hence, the ERCE [63] was designed as a prospective, observational, multilingual, multicenter, online clinical registry for patients with probable or proven CE, serving as a crucial part of the HERACLES joint effort.

##### Objectives and Results

Ultrasound screening of CE within the Eastern European population and the ERCE register.

The study approached the subject from two angles: firstly, it used an ultrasound screening protocol to assess the prevalence and burden of CE in underserved rural regions with a substantial sheep farming industry (Bulgaria, Romania, and Turkey); and second, it constructed a comprehensive ERCE database that included Bulgaria, Romania, and Turkey [15,58,60,63,64,65]. 

One of the project’s triumphs was a cross-sectional ultrasound-based study [15], for which volunteers were recruited from fifty rural towns in Turkey, Bulgaria, and Romania. The communities chosen were located in provinces where the frequency of CE in regional hospitals was comparable to the national average. During the screening period, lesions were detected and categorized using a modified WHO approach. After adjusting for age and gender, the prevalence rates of abdominal CE were determined using direct standardization, with the rural population of each country serving as the reference [15]. A cumulative total of 24,693 individuals were screened; of them, 24,687 underwent ultrasonography examinations. The estimated prevalence of abdominal CE in Bulgaria was 0.41% (0.29–0.58), in Romania, it was 0.41% (0.26–0.65), and in Turkey, it was 0.59% (0.26–1.85) [15,60].

The creation of ERCE was driven by the need for a surveillance tool that Is better suited for monitoring human CE. It takes into consideration unique characteristics of the infection, such as its chronic nature and common practice outpatient case management. Additionally, ERCE aimed to establish a case database that allows for a prospective assessment of both the natural progression of individual cysts, as well as any changes resulting from treatment [60,63]. The study investigated the efficacy of a stage-specific treatment method, the incidence of side effects, the rate of recurrence, and the related costs of treating CE infection in a prospective and systematic manner [63]. 

Data gathered under the ERCE initiative have been supplying national authorities and international entities such as the European Centre for Disease Prevention and Control (ECDC) with the means to recognize the extent of the issue by documenting cases that may otherwise go unaccounted for in systems like hospital discharge records.

2.New molecular-based tools for detection, diagnosis, and follow-up of CE.

A key objective was to establish and improve upon a consistent method for the serological identification of CE in both human and animal subjects [60]. A systematic approach was taken to examine the serodiagnosis and serological monitoring of CE. A sample repository was established and connected to databases containing comprehensive clinical and epidemiological data. This ensured the accurate validation of newly identified recombinant antigens and their corresponding antibodies [60,66,67,68,69,70,71]. 

Initially, the experiment looked at the variables that impact the serological response in individuals with hepatic CE using commercially accessible ELISA and IHA assays that are regularly utilized in parasitology laboratories [72]. It was concluded that the serological responses that are evaluated by these tests are influenced by factors such as the activity, size, and quantity of CE cysts, as well as the time of serum collection relative to the treatment. 

The second experiment assessed the diagnostic precision of three commercially accessible rapid diagnostic tests (RDTs) in detecting hepatic CE [73]. One test in particular (the VIRapid test, Vircell, Spain) was demonstrated to perform better when compared to the other evaluated kits; its results were comparable to those of the control test (ELISA). However, it is important to note that all the tests exhibited low sensitivity for inactive cysts [73].

The process of selecting, testing, and producing recombinant antigens was undertaken to ensure their compatibility with point-of-care/low-cost (POC-LOC) devices, commonly referred to as commercial diagnostic assays. Experiments in immunochromatography were conducted for the antigen 2B2t, leading to the development of a prototype that has undergone preliminary validation, but this did not provide sufficient evidence to support its utilization [60,67]. Other options, such as AgB, Ag5, and other potential antigenic markers, were also tested [70].

3.Host–parasite interplay.

The understanding of the underlying molecular pathogenicity mechanisms contributed to the identification of diagnostic and prognostic biomarkers to assist in the care of infected individuals. Additionally, it was critical to determine if the genetic diversity of hosts has led to the observed clinical heterogeneity among study subjects [60,71].

The investigation of haplotypes, genotypes, and species [71] was followed by microRNA arrays [74]. The expression of eight specific microRNAs (let-7g-5p, let-7a-5p, miR-26a-5p, miR-26b-5p, miR-195-5p, miR16-5p, miR-30c-5p, and miR-223-3p) was found to be increased in cases where active cysts were present. It was concluded that host microRNAs play a role in regulating the immune response against *E. granulosus* and/or in the development of hydatid cysts [74].

The discovery of accessible circulating biomarkers could allow for the creation of a diagnostic test that could considerably improve CE diagnostic rates [60]. The focus was set on exosomes—a type of extracellular vesicle that play a crucial role in intercellular communication, particularly in immune system responses. Following proteomic assays, the proteins were divided into ”true” biomarkers and potential biomarkers that need to be further studied [75]. Research identified the Src family kinases (Src and Lyn) as potential indicators of active CE, based on their presence in distinct plasma pools. An association between TGF-β in active CE and Cdc42 in inactive CE was also identified.

4.The increasing bioavailability of albendazole and a new enantiomeric drug synthesis.

The primary goals encompassed the synthesis of novel racemic and enantiomeric pure medicines derived from albendazole (ABZ), as well as the evaluation of their efficacy against *E. granulosus* through in vivo experimentation [60,76,77,78,79]. Salts of compounds with a benzimidazole structure were patented [64] and included in a trial with a murine model [79] and two clinical trials [22,60]. The novel salt formulations of ABZ, ricobendazole (RBZ), and the enantiomer RBZ showed promising results and call for future research. The patent includes more antihelmintic compounds: fenbendazole, triclabendazole, flubendazole, and others [77]. These substances have not undergone comprehensive investigation and provide a backbone for future research endeavors.

5.Training and dissemination of information.

##### Correlation with the WHO Road Map

The HERACLES project effectively targets a substantial number of the suggestions that are put forth by the WHO. Noteworthy among these recommendations are:The establishment of an international patient surveillance network for cystic echinococcosis (ERCE);The implementation of screening programs in rural parts of the countries that are at risk, as well as the execution of a study aimed at determining the prevalence rates in these specific regions;Reporting to both national and international authorities tasked with the responsibility to control and prevent infectious diseases, and thereafter engaging in collaborative efforts to formulate recommendations that are targeted at managing the endemic situation;Improving diagnostic techniques by identifying novel biomarkers that can be employed in the molecular diagnosis of CE.

Additionally, it acknowledged, evaluated, and patented novel antihelmintic compounds.

#### 3.4.2. The mEmE Project

Project mEmE (Multi-centre study on *Echinococcus multilocularis* and *Echinococcus granulosus* sensu lato in Europe: development and harmonization of diagnostic methods in the food chain), carried out between 2013 and 2018, was a collaborative initiative between multiple international centers. Its primary objective was to address research gaps that were identified by international agencies tasked with the control of zoonotic parasites, specifically *E. multilocularis* (Em) and *E. granulosus* sensu lato (Eg s.l.) [80,81,82]. The research was focused on epidemiological and molecular studies to later establish clear guidelines.

The consortium for this project included several countries (Italy, Germany, France, the Netherlands, Poland, Denmark, Portugal, Estonia, and Latvia) and some external partners (Ireland, Switzerland, Norway, and Pakistan) [80]. The approach of the project included a veterinary point of view on the topic, thus reinforcing the need for collaboration between specialists towards the containment of the NTD.

##### Objectives and Results

Standard operating procedures (SOPs) for sampling.

One of the early achievements of the project was the delivery of standard operating procedures (SOPs) for sampling [3,49,83]. The samples that were gathered encompass a wide range of components within the food chain: excrements and digestive systems of the primary carnivorous hosts, vegetables intended for human consumption, and cysts found in the intermediate cattle hosts.

The SOPs have demonstrated significant utility within the project and have the potential to become a standard methodology for sample collection in the years to come.

2.Validation of multiplex PCRs.

Two methods were chosen for this project [80,82], (1) an assay [84] that used a single-tube multiplex polymerase chain reaction (PCR) method that enables the differentiation of *Echinococcus* species and genotypes; (2) and another assay [85] that used a single-tube multiplex PCR method to identify eggs of *E. granulosus*, *E. multilocularis*, and the *Taenia* genus by targeting the definitive host.

3.Sequencing of samples using RSE and NGS.

The method used DNA enrichment techniques with parasite-specific DNA capture probes and magnets to obtain parasite DNA from complex biological materials such as feces or raw vegetables [82,86]. Furthermore, the captured fragments were subjected to high-throughput long- and short-read sequencing technologies to obtain their comprehensive characterization.

4.The prevalence of Em/Eg in dogs from selected geographical areas [87].5.A quantitative assessment of the impact of human CE in Europe.

The mean annual incidence for the period of 1997–2020 was 0.64 per 100,000 individuals across European nations and 0.50 per 100,000 individuals specifically in EU countries [82]. However, certain regions in Europe have been identified as high-endemicity areas for cystic echinococcosis [3,20], with reported cases ranging from 1 to 5 per 100,000 individuals. These areas included Albania (2.25 per 100,000), Bosnia and Herzegovina (1.00 per 100,000), Bulgaria (5.33 per 100,000), Italy (1.21 per 100,000), Moldova (4.65 per 100,000), North Macedonia (1.08 per 100,000), and Romania (2.16 per 100,000). 

While exact patterns differed nationally, there was evidence of an overall drop in frequency. Most endemic nations in Southern and Eastern Europe, where the illness had historically exhibited a high prevalence, had documented a decline in the incidence of CE [62,82]. In contrast, there has been an unforeseen rise in the majority of non-endemic Northern (Scandinavia) and Western European nations, as well as Baltic countries [80,82]. Based on the most recent data and trends for European countries, it was noted that the Balkan region should be regarded as the prevailing epicenter of cystic echinococcosis within Europe.

6.Training and Dissemination of information.

##### Correlation with WHO Road Map

The mEmE initiative addressed WHO’s appeal to modify the status of CE by employing a range of approaches:The establishment of SOPs on sample collection in cases of CE suspicion, particularly in animals, which has played a significant role in contributing to disease control and prevention efforts;The basis for an epidemiological study within the present context;Confirmation of the efficacy of molecular diagnostic techniques;A wide distribution of project findings, targeting both the general public and experts in human and veterinary disciplines.

One of the most important epidemiological findings was that the geographical focal point (increased incidence rates) was not in Southern or Eastern Europe but rather in the Baltic/Scandinavian area. The need to standardize an operational protocol is heightened, as it plays a critical role in attaining the targets that have been established by the WHO.

A relationship summary between the HERACLES project, the mEmE project, and the WHO road map for neglected tropical diseases 2021–2030 can be found in Table 2.

## 4. Discussion

The World Health Organization (WHO) places significant emphasis on the need for preventative and control measures in its official publication. Accurately and promptly identifying [1,14,18] epidemic foci is one of the significant challenges that the professionals face. The absence of symptoms that is encountered in infected canids and livestock makes CE monitoring in animals difficult; communities and local veterinary services might not currently acknowledge or assign importance to surveillance.

Research initiatives such as HERACLES and MEmE serve as the basis for a comprehensive database [63] that encompasses up-to-date epidemiological data on the state of the disease in Europe. They represent a significant step towards adding a geographical and, implicitly, an economic dimension to the medical context for Europe in particular, and for Asia, the Middle East, and North Africa in general.

### 4.1. Prospects for Prevention and Control

Following Table 1, where the pooled prevalence across Asia was 0.0177, numerous studies found substantially higher rates of CE in certain regions, such as rural Mongolia (2.0% to 13.1%) [88], the Fars province of Southern Iran (8.73%) [89], and rural Zahedan, Iran (4%) [90]. The same holds true for South America, where the pooled prevalence was 0.0342, but higher prevalence rates were found in parts of Potos Department, Bolivia (4,3%) [91,92], and La Rioja Province, Argentina [93], emphasizing the need for more effective surveillance and control methods worldwide.

The vaccination of sheep with the *E. granulosus* recombinant antigen EG95 vaccine seems promising in preventing and controlling CE [41,42,94,95]. This vaccine is now being manufactured commercially and is registered in China and Argentina. A pilot study conducted in Argentina [94] has highlighted the significant advantages of the vaccine in mitigating the transmission of disease among sheep, a promising advancement in the efforts to prevent echinococcosis. These interventions are part of the One Health approach [2,81,96,97], emphasizing the interconnectedness of human and animal health.

Several countries, including Morocco and Mongolia, have undertaken initiatives aiming at decentralizing diagnostic and treatment approaches and adopting early detection. In Bulgaria, Romania, and Turkey, a cross-sectional study [15] has highlighted the challenges of understanding the true burden of CE, as many cases remain asymptomatic, without proper medical diagnosis and treatment.

In other parts of the world, such as Mongolia, the WHO has initiated a national action plan for the control of echinococcosis, promoting cross-sectoral collaboration [98,99]. Similarly, China is integrating echinococcosis prevention, control, and treatment into its economic and development plans, with a focus on the Central Asian Republics [100].

In the Americas, a manual for cystic echinococcosis control was published by the Pan American Health Organization (PAHO) and WHO Regional Office for the Americas (OPS) in 2017 [101].

WHO collaborates with the World Organization for Animal Health (WOAH) [3] to support the development of echinococcosis control programs that include animal interventions. These programs aim to reduce the disease burden and safeguard the food value chain. WHO also assists countries in developing and implementing pilot projects for effective CE control strategies.

### 4.2. Limitations of the Study

This study faces several limitations that impede a comprehensive understanding of the current global distribution of the disease. One such limitation lies in the study’s reliance on a relatively small sample size within the broader spectrum of available research, potentially constraining the generalizability of the findings. Furthermore, despite using our chosen keywords and running searches across multiple databases, it is possible that studies with sizable participant counts were still overlooked.

The absence of epidemiological data significantly hampers the delineation of the disease’s profile worldwide. This shortfall partly arises from underreporting to responsible organizations. Furthermore, even when data are available, these entities of various backgrounds may need different levels of assistance in assimilating data and providing clear formats to follow. Challenges such as language barriers and differences in data formats further exacerbate this issue.

Variations in screening methods and study inclusion criteria contribute to discrepancies between sources. For example, some researchers recruit suspected cystic echinococcosis patients during hospitalization, while others employ ultrasound-based screening protocols. These methodological disparities likely account for the significant divergences in epidemiological data across multiple studies.

Few studies holistically address the issue of cystic echinococcosis, encompassing screening and diagnosis, treatment, control measures for animal-to-human transmission, and preventive strategies for at-risk populations. The trend is to discuss confounding factors independently, focusing solely on prevalence OR incidence in animals OR humans, leading to interventions primarily targeting one or the other.

## 5. Conclusions

There are multiple unexplored avenues of research that possess the potential to make substantial contributions to the control and prevention of cystic echinococcosis. The EG95 recombinant antigen vaccine for sheep can be considered a prime example of a significant step. Additional efforts are currently being undertaken to expand the use of this vaccine worldwide.

Furthermore, the improved accessibility of benzimidazole therapies is a continuous endeavor. Initiatives are currently in progress to examine different strategies that are aimed at improving the accessibility and affordability of these drugs, hence boosting their availability to affected populations.

The management of echinococcosis relies heavily on the collaboration between professionals, including microbiologists, infectious diseases specialists, epidemiologists, public health specialists, veterinary physicians, agricultural workers, ecologists, wildlife experts, and other experts. This enables the improvement of monitoring techniques, the progression of molecular diagnostics, the cultivation of a deeper understanding of the disease at the host-parasite level, and the development of more effective intervention options.

To conclude, encouragement of interdisciplinary collaborations facilitated the development of standardized protocols pertaining to the collection of samples, diagnostic methodologies, and new treatment strategies. Collaboration between experts is critical not only for filling current knowledge gaps about the disease, but also for developing strategies for its containment, making progress in reducing underreporting, and ultimately alleviating the disease’s effects on human and livestock health.

## Figures and Tables

**Figure 1 tropicalmed-09-00036-f001:**
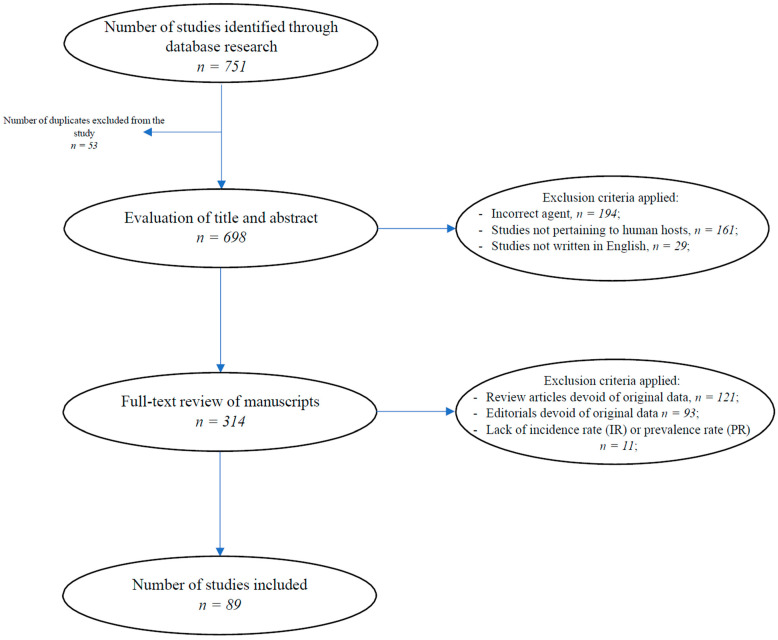
The selection process for the data included in the study.

**Table 1 tropicalmed-09-00036-t001:** Analysis of the data included in the most recent studies addressing CE worldwide, as reported by *Brunetti E. and Tamarozzi F*. [40].

Continent	Information Regarding Data		Mean Prevalence
Africa	Number of studies included	4	0.0114 (CI 95%, 0.0089–0.0139)
	Time frame of the included studies (publishing date)	2013–2022	
	Number of patients included	7002	
	Number of identified CE patients	80	
South America	Number of studies included	6	0.0342 (CI 95%, 0.0296–0.0389)
	Time frame of the included studies (publishing date)	2014–2022	
	Number of patients included	5845	
	Number of identified CE patients	200	
Asia	Number of studies included	7	0.0177 (CI 95%, 0.0169–0.0185)
	Time frame of the included studies (publishing date)	2011–2022	
	Number of patients included	98,751	
	Number of identified CE patients	1747	
Europe	Number of studies included	3	0.0027 (CI 95%, 0.0020–0.0035)
	Time frame of the included studies (publishing date)	2018–2020	
	Number of patients included	18,495	
	Number of identified CE patients	50	
Total	Number of patients included	130,093	0.0160 (CI 95% 0.0153–0.0166)
	Number of identified CE patients	2077	

**Table 2 tropicalmed-09-00036-t002:** Correlation between the achievements of the HERACLES project, the mEmE project, and the WHO road map for neglected tropical diseases 2021–2030. The WHO document encompasses several more recommendations; however, not all topics were covered within the scope of the two mentioned projects. Therefore, not all the WHO recommendations have been included in the summary.

WHO Road Map Required Actions		HERACLES Project	mEmE Project
Scientific understanding	Map health and economic burden	International patient surveillance network for cystic echinococcosis (ERCE)	A systematic review focused on the quantitative assessment of the impact of CE in Europe [82]
	Estimate prevalence in sheep and other relevant livestock	-	Prevalence of Em/Eg in dogs from selected geographical areas
	Research to quantify resources needed to control the disease	56 publications with implications on how to better understand and consequently control the disease	18 publications, including epidemiological trends, diagnostic methods, and others
Diagnostics	Define target product profile and develop optimal diagnostic for humans	The discovery of new biomarkers, with the possibility of further research on the topic Evaluation of commercial serological tests and the identification of their limits Efforts towards implementation of POC-LOC devices	-
Effective intervention	Conduct efficacy trials in humans to understand optimal treatment courses of albendazole	Novel antihelmintic compounds have been discovered, evaluated, and patented The new molecules were defined as salts with a benzimidazole structure, e.g., albendazole	-
Operational and normative guidance	Develop guidelines for implementation of optimized prevention and control methods	-	The establishment of SOPs on sample collection in cases of CE suspicion, particularly for dogs, sheep, foxes, and others
Planning governance and program implementation	Implement systematic use of population ultrasound screening for early diagnosis	A cross-sectional ultrasound-based study in rural parts of the countries involved, aimed at determining the prevalence rates in these specific regions	-
	Mandate segregated reporting of CE and AE in all endemic countries	International patient surveillance network for cystic echinococcosis (ERCE)	-
Capacity and awareness building	Develop training courses for medical personnel on diagnosis and clinical management of CE and AE in rural areas of affected countries	The projects have successfully disseminated the acquired information among the local communities of partner nations through several channels: (1) Publishing their research to communicate novel discoveries to the scientific community; (2) Implementing public health initiatives aimed at healthcare professionals, utilizing research outcomes as a basis; (3) Disseminating information to the general population through various means, such as social events, brochures, or journals.	
	Provide community education based on the local values to improve the effectiveness of interventions		

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
