# Peer review of "Cystic Echinococcosis in the Early 2020s: A Review"

_tropicalmed, 2024, doi:10.3390/tropicalmed9020036_

Round 1

Reviewer 1 Report

Comments and Suggestions for Authors

Major comments:

- Why does the research describe the use of several databases, but only present results from PUBMED/Medline? This needs to be reviewed in the methodology and if there is no data beyond PUBMED/Medline, the authors need to redo the study with at least three DATABASES: Web of Science; Science Direct; and PUBMED, apply the inclusion and exclusion criteria and include data that may have been lost due to the use of just one database.

- Line 101: Were animal studies included or not?

- Considering that 148 reviews were found by the authors, the objective of this review, as well as its need, are not clear in the text. Why is this review important and how does it differ from others? Either this becomes clear, or it should not even be published. Discuss in the Introduction.

- Why is the diagnosis not listed in the guidelines (item 3.3)?

- Line 490-500: This text is not a discussion, as it does not promote dialogue between literature.

Minor comments:

- Keywords are many and not entirely suitable. They do not need to repeat the review search terms. Suggestion: Echinococcus granulosus; Fooodborne Disease; One Health; Zoonosis.

- Line 59: Replace "(family Taeniidae)" with "(Cestoda: Taeniidae)".

- Sensu lato and sensu stricto should not be in italics, as should their abbreviations.

- Line 111: Replace "Echinococcus multilocularis" with "E. multilocularis" and write in italics.

- In item 3.1 Taxonomy, scientific names already used must have the genus name abbreviated.

- Presentation of data is confusing: materials and methods are not followed. The selected manuscripts (52) are not clearly discussed in the results. The impression one gets is that they were only used to estimate prevalence and incidence. Furthermore, as complex as it may be, citing the use of national guidelines only in English is scientifically exclusionary, because if there is any guidance in countries outside of Europe and North America, are these data not scientifically relevant? This becomes biased, especially considering prevalence rates in countries in Asia, South America and Africa. As a reader, I wonder if there is something relevant being done in these countries, and I wonder if this is reflected in the approach given to two European projects. I believe that there is a lot of work to be done by the authors, or the perspective and objectives of the study are not that efficient.

Author Response

Esteemed Reviewer,

Herein we, the Authors, will respond to the commentaries provided in the first round of review of our manuscript. We hope that our response is consistent with the Reviewer’s expectations and that it answers the concerns raised prior to this.

For ease of reference, we will be including the Reviewer’s notes using quotation marks and italics, with our response detailed below. Quotations from the updated manuscript are instead highlighted using bold.

As our manuscript has undergone some changes since its initial submission, the location and positioning of some lines of text have changed; as such, for our response, we will be referencing their position in the updated PDF manuscript.

Please refer to the PDF version of the manuscript for the updated line numbers.

          Please note that following the suggestion of the reviewers, Supplementary File: ”Table_S1”, became ”Table_S3”.

“Major comments:

- Why does the research describe the use of several databases, but only present results from PUBMED/Medline? This needs to be reviewed in the methodology and if there is no data beyond PUBMED/Medline, the authors need to redo the study with at least three DATABASES: Web of Science; Science Direct; and PUBMED, apply the inclusion and exclusion criteria and include data that may have been lost due to the use of just one database.”

We thank the Reviewer for pointing out the shortcomings in our Methodology description. In our text, we have unwillingly highlighted the use of MeSH search in PubMed more than the other queried databases. In fact, MeSH search was used in PubMed and the same keywords were used for all the other databases. To that purpose, we have updated the text in the manuscript to read Lines 91-93: In PubMed, The Medical Subject Headings (MeSH) technique was used, and parallel strategies employing identical keywords were used in the other available databases (Web of Science, Cochrane, ScienceDirect, PubMed Central, and Google Scholar).” and Lines 110-111: “The total number of research articles obtained through search of PUBMED/Medline database searches of the titles and abstracts was 751.”

“- Line 101: Were animal studies included or not?”

          The Reviewer asks a legitimate question about the inclusion of studies pertaining to animals. These studies are outside the scope of our review.  We have updated the text in the manuscript to read “(i.e., human and animal prevalence rates incidence rates (IR) and prevalence rates (PR)), treatment options, and new research contributing to the control and prevention of the disease (i.e., new biomarkers used for diagnosis, etc.)”. All remaining information regarding the incidence and prevalence rates in animals are to be taken into consideration only in relation to human incidence and prevalence rates, such as in Table S1. We thank the Reviewer for pointing out this shortcoming and for its contribution to improving the manuscript.

“- Considering that 148 reviews were found by the authors, the objective of this review, as well as its need, are not clear in the text. Why is this review important and how does it differ from others? Either this becomes clear, or it should not even be published. Discuss in the Introduction.”

Although the Reviewer correctly notes that there are 148 reviews available in the literature, we believe that an updated perspective—especially one that emphasizes the most recent epidemiological data and novel findings and approaches adopted by various nations to address the issue—is not only pertinent but also essential for the scientific community to advance, given the disease's neglected status as stated by the World Health Organisation in their most recent document regarding the topic (the Roadmap). These considerations are relevant not just in light of a shortage in reporting in some regions of the world, but also because they stand as grounds for dialogue between researchers. For further details, please refer to the new Supplementary Files (table S1 and S2).

“- Why is the diagnosis not listed in the guidelines (item 3.3)?”

The Reviewer has aptly pointed out the absence of a diagnosis sub-section in the Current Guidelines section of the review (item 3.3). The authors concluded that no significant difference was discovered after consulting the relevant guidelines listed in the Materials and Methods section. As such, we have summarized all diagnostic methodologies in the Introduction (Lines 35-57).

“- Line 490-500: This text is not a discussion, as it does not promote dialogue between literature.”

We thank the Reviewer for their suggestion and concur that the referenced text is not suitable for the Discussion section of our manuscript. As such, we have removed it entirely.

“Minor comments:

- Keywords are many and not entirely suitable. They do not need to repeat the review search terms. Suggestion: Echinococcus granulosus; Fooodborne Disease; One Health; Zoonosis.

- Line 59: Replace "(family Taeniidae)" with "(Cestoda: Taeniidae)".

- Sensu lato and sensu stricto should not be in italics, as should their abbreviations.

- Line 111: Replace "Echinococcus multilocularis" with "E. multilocularis" and write in italics.

- In item 3.1 Taxonomy, scientific names already used must have the genus name abbreviated.”

We thank the Reviewer for their suggestions and have adapted the text accordingly.

Additionally, a manual review of the manuscript indicated other instances where italics and abbreviations were needed.

“- Presentation of data is confusing: materials and methods are not followed. The selected manuscripts (52) are not clearly discussed in the results. The impression one gets is that they were only used to estimate prevalence and incidence. Furthermore, as complex as it may be, citing the use of national guidelines only in English is scientifically exclusionary, because if there is any guidance in countries outside of Europe and North America, are these data not scientifically relevant? This becomes biased, especially considering prevalence rates in countries in Asia, South America and Africa.”

We thank the Reviewer for bringing the matter of including information on national guidelines and studies from outside the European region. To this end, we have added a Supplementary File (Table S1) to represent more clearly the national guidelines in locations with high prevalence rates in South America, Asia, and Africa, we have found that they follow the recommendations of the World Health Organization. Furthermore, we strive to include in our manuscript the relevant work of all of our fellow colleagues from around the world, and we regret any omissions we may have made in this regard.

“As a reader, I wonder if there is something relevant being done in these countries, and I wonder if this is reflected in the approach given to two European projects. I believe that there is a lot of work to be done by the authors, or the perspective and objectives of the study are not that efficient.”

The Reviewer raises concerns about the presentation of Results, in the context of the stated materials and methods. We have modified this section accordingly and have included in both this section and the Discussion section a series of studies which we have found to offer a more comprehensive view on the matter.

We would also like to point out that Table 1 contains an overview of incidence and prevalence rates from recent research, as well as a more complete description divided by continents and nations in the Supplementary File (please see Table S3). Although the HERACLES and MEmE projects occupy a significant portion of the discussed material, their identity as European projects is not the focus of our review, as mentioned in the Introduction, but rather that these are the projects we have found to most align with the WHO Roadmap.

To add to this, we have found several non-European studies in our search that were not eligible for inclusion in our review because of the use of outdated diagnosis criteria, lack of clarity in the etiological agent involved (no distinction between E. granulosus and E. multilocularis), and other factors related to our review’s inclusion and exclusion criteria [1-7], which we have cited below.

Even with our repeated searches using the mentioned keywords, studies with sizable participant counts could have still been overlooked. One such instance was identified in the case of a 2019 study by Li B et al. [8] which was identified when studying the work of Brunetti and Tamarozzi (citation 40 in the updated manuscript). We have edited the limitations subsection of the Discussion section accordingly.

We would respectfully wish to ask the Reviewer to share their expertise in identifying seminal works that may have been overlooked and would contribute to the improvement of the manuscript.

We gratefully acknowledge the Reviewer's criticisms and overall evaluation of our manuscript, and we hope that the answer provided herein, as well as the amendments made to the paper meet the Reviewer's expectations.

Respectfully,

The Authors

References

  1. Khan, A.; Ahmed, H.; Khan, H.; Saleem, S.; Simsek, S.; Brunetti, E.; Afzal, M.S.; Manciulli, T.; Budke, C.M. Cystic Echinococcosis in Pakistan: A Review of Reported Cases, Diagnosis, and Management. Acta Tropica 2020, 212, 105709, doi:10.1016/j.actatropica.2020.105709.
  2. Colpani, A.; Achilova, O.; D’Alessandro, G.L.; Budke, C.; Mariconti, M.; Muratov, T.; Vola, A.; Mamedov, A.; Giordani, M.T.; Suvonkulov, U.; et al. Ultrasound-Based Prevalence of Cystic Echinococcosis in the Samarkand Region of Uzbekistan: Results from a Field Survey. Am J Trop Med Hyg 2023, 109, 153–158, doi:10.4269/ajtmh.22-0376.
  3. Feng, X.; Qi, X.; Yang, L.; Duan, X.; Fang, B.; Gongsang, Q.; Bartholomot, B.; Vuitton, D.A.; Wen, H.; Craig, P.S. Human Cystic and Alveolar Echinococcosis in the Tibet Autonomous Region (TAR), China. J Helminthol 2015, 89, 671–679, doi:10.1017/S0022149X15000656.
  4. Luo, A.; Wang, H.; Li, J.; Wu, H.; Yang, F.; Fang, P. Epidemic Factors and Control of Hepatic Echinococcosis in Qinghai Province. J. Huazhong Univ. Sci. Technol. [Med. Sci.] 2014, 34, 142–145, doi:10.1007/s11596-014-1246-8.
  5. Paternoster, G.; Boo, G.; Flury, R.; Raimkulov, K.M.; Minbaeva, G.; Usubalieva, J.; Bondarenko, M.; Müllhaupt, B.; Deplazes, P.; Furrer, R.; et al. Association between Environmental and Climatic Risk Factors and the Spatial Distribution of Cystic and Alveolar Echinococcosis in Kyrgyzstan. PLoS Negl Trop Dis 2021, 15, e0009498, doi:10.1371/journal.pntd.0009498.
  6. Acosta-Jamett, G.; Hernández, F.A.; Castro, N.; Tamarozzi, F.; Uchiumi, L.; Salvitti, J.C.; Cueva, M.; Casulli, A. Prevalence Rate and Risk Factors of Human Cystic Echinococcosis: A Cross-Sectional, Community-Based, Abdominal Ultrasound Study in Rural and Urban North-Central Chile. PLoS Negl Trop Dis 2022, 16, e0010280, doi:10.1371/journal.pntd.0010280.
  7. Reyes, M.M.; Taramona, C.P.; Saire-Mendoza, M.; Gavidia, C.M.; Barron, E.; Boufana, B.; Craig, P.S.; Tello, L.; Garcia, H.H.; Santivañez, S.J. Human and Canine Echinococcosis Infection in Informal, Unlicensed Abattoirs in Lima, Peru. PLoS Negl Trop Dis 2012, 6, e1462, doi:10.1371/journal.pntd.0001462.
  8. Li, B.; Quzhen, G.; Xue, C.-Z.; Han, S.; Chen, W.-Q.; Yan, X.-L.; Li, Z.-J.; Quick, M.L.; Huang, Y.; Xiao, N.; et al. Epidemiological Survey of Echinococcosis in Tibet Autonomous Region of China. Infect Dis Poverty 2019, 8, 29, doi:10.1186/s40249-019-0537-5.

Reviewer 2 Report

Comments and Suggestions for Authors

The review "Cystic Echinococcosis in the early 2020s: a review" is well written and very detailed. 

Minor changes suggested: 

1) When is the start of data collection, what time period? from XY-11/2023

2) The Keywords are "Echinococcus granulosus", "E. granulosus", "echinococcosis", "cystic echinococcosis", other keywords such as "hydatid disease" were also taken into account?

3) Line 111: Echinococcus multilocularis is not italicized 

4) Figure 1: The presentation at the beginning is confusing, the excluded papers n=53 should be on the left with a horizontal arrow and the 2nd vertical field should read "Evaluation of title and abstract n=698". 

5) Table 1: There should be a space, colored separation or a dividing line between the individual regions, otherwise it is too confusing. 

6) Line 253: "Chemotherapy" The dosage is missing and alternatives to albendazole and mebendazole are missing such as praziquantel.

7) Line 264: The topic of "chemotherapy before surgery" is controversially discussed, when should chemotherapy be started before surgery, 1 month, one week or on the day of surgery? Is there more information? Only one publication (?) Complication rates? This should be analyzed in more detail.

Author Response

Esteemed Reviewer,

Herein we, the Authors, will respond to the commentaries provided in the initial review of our manuscript. We hope that our response is consistent with the Reviewer’s expectations and that it answers the concerns raised prior to this.

For ease of reference, we will be including the Reviewer’s notes using quotation marks and italics, with our response detailed below. Quotations from the updated manuscript are instead highlighted using bold.

As our manuscript has undergone some changes since its initial submission, the location and positioning of some lines of text have changed; as such, for our response we will be referencing their position in the updated manuscript.

Please refer to the PDF version of the manuscript for the updated line numbers.

Please note that following the suggestion of the reviewers, Supplementary File: ”Table_S1”, became ”Table_S3”.

“Minor changes suggested:

 1) When is the start of data collection, what time period? from XY-11/2023”

No definitive start period was selected for data collection; all articles returned by our database query were included in the study.

“2) The Keywords are "Echinococcus granulosus", "E. granulosus", "echinococcosis", "cystic echinococcosis", other keywords such as "hydatid disease" were also taken into account?”

We thank the Reviewer for their suggestion. For simplicity, we did not include variations of the main keywords mentioned in the article, as their inclusion did not modify our data query. In order for this to be made clearer, we have modified lines 98-99 in the updated manuscript:  "Other common terms associated with CE (i.e. "hydatid disease”, "hydatid cyst", etc.) were also included in the database search, but did not yield any additional findings."

 “3) Line 111: Echinococcus multilocularis is not italicized”

 We thank the Reviewer for their suggestion. We have adapted the text accordingly.

“4) Figure 1: The presentation at the beginning is confusing, the excluded papers n=53 should be on the left with a horizontal arrow and the 2nd vertical field should read "Evaluation of title and abstract n=698".“

We thank the Reviewer for their observation. The image was changed to reflect their thoughtful recommendation.

“5) Table 1: There should be a space, colored separation or a dividing line between the individual regions, otherwise it is too confusing.”

          We agree with the Reviewer’s insight and have added a dividing line between the regions for better readability.

“6) Line 253: "Chemotherapy" The dosage is missing and alternatives to albendazole and mebendazole are missing such as praziquantel.

 7) Line 264: The topic of "chemotherapy before surgery" is controversially discussed, when should chemotherapy be started before surgery, 1 month, one week or on the day of surgery? Is there more information? Only one publication (?) Complication rates? This should be analyzed in more detail.”

          Concerning albendazole use in asymptomatic patients, we have made changes in the Chemotherapy and “Watch-and-wait” approach portions of the 3.3.2 Treatment subsection, particularly lines 285-288, 297-301 and 316-319 to add more insight into both this matter and the matter of albendazole and praziquantel use in patients in general. To this end, we consider that clinical practice varies depending on multiple factors, which we have mentioned in the full text manuscript, lines: 268-271.

          The best choice for Albendazole (used before and/or after surgery, or only after surgery) was discussed with different variations by different authors, for different patients, or different localizations.

          Please be assured that we know that you know all these aspects and we are not trying to present this data/information as if we have a better knowledge than other colleagues. We have chosen to mention only a part of all the possibilities because the way you put the problem is pertinent, exciting, worth discussing. But either considering the articles read, or considering the experience of the clinic in Bucharest (1994 is the year of inauguration), we cannot choose a variant that covers all possibilities.

We sincerely thank the Reviewer for their comments and overall evaluation of our manuscript, and we hope that the response herein, as well as the changes made to the manuscript meet the Reviewer’s expectations.

Respectfully,

The Authors

Reviewer 3 Report

Comments and Suggestions for Authors

I cannot fully understand the objective of this work.

In the introduction, special reference is made to the diagnosis and clinical presentation in people.

From epidemiology, only mention is made of the occurrence in Europe.

The objectives of the bibliographic review are not specified.

Is it about all hosts? Is it about clinical and diagnosis in people? Does it reach everyone or just Europe? Are you interested in control?

This must be clarified to understand the work

Then the work advances on certain search and inclusion criteria, not related to the introduction text.

“The inclusion criteria for our research included manuscripts that offered relevant in-

formation regarding cystic echinococcosis, such as taxonomic changes, epidemiological

data (i.e., human and animal prevalence rates and incidence rates), treatment options, and new research contributing to the control and prevention of the disease (i.e., new biomarkers used for diagnosis, etc.)”.  

The inclusion criteria are not clear. For example. Real control experiences with different strategies do not appear as eligible.

From this there is a strong and surprising omission of published control experiences, of the use of albendazole in asymptomatic patients and of the systematic use of field ultrasonography. This is associated with a selected bibliography that appears very biased towards certain authors with the omission of others.

It should be better explained why these thematic axes were used and based on what criteria some publications are admitted.

Perhaps this would be clearer if the work is directed ONLY to Europe, WHICH would require an overall adjustment of the work.

This seems very important as the work is based on an exhaustive review of databases of published works.

Author Response

Esteemed Reviewer,

Herein we, the Authors, will respond to the commentaries provided in the initial review of our manuscript. We hope that our response is consistent with the Reviewer’s expectations and that it answers the concerns raised prior to this.

For ease of reference, we will be including the Reviewer’s notes using quotation marks and italics, with our response detailed below. Quotations from the updated manuscript are instead highlighted using bold.

Please refer to the PDF version of the manuscript for the updated line numbers.

Please note that following the suggestion of the reviewers, Supplementary File: ”Table_S1”, became ”Table_S3”.

As our manuscript has undergone some changes since its initial submission, the location and positioning of some lines of text have changed; as such, for our response we will be referencing their position in the updated manuscript.

“In the introduction, special reference is made to the diagnosis and clinical presentation in people.

From epidemiology, only mention is made of the occurrence in Europe.”

          The Reviewer raises concerns about the presentation of Results, in the light of the indicated materials and methods. We have modified this section accordingly and have included a series of studies which we have found to offer a more comprehensive view on the matter in both this section and the Discussion section.

         We would also like to point out that Table 1 contains an overview of incidence and prevalence rates from recent research, as well as a more complete description divided by continents and nations in the Supplementary File (please see Table S3). Although the HERACLES and MEmE projects occupy a significant portion of the discussed material, their identity as European projects is not the focus of our review, as mentioned in the Introduction, but rather that these are the projects we have found to most align with the WHO Roadmap.

         To add to this, we have found several non-European studies in our search that were not eligible for inclusion in our review because of the use of outdated diagnosis criteria, lack of clarity in the etiological agent involved (no distinction between E. granulosus and E. multilocularis), and other factors related to our review’s inclusion and exclusion criteria [1-7], which we have cited below.

       Even with our repeated searches using the mentioned keywords, studies with sizable participant counts could have still been overlooked. One such instance was identified in the case of a 2019 study by Li B et al. [8] which was identified when studying the work of Brunetti and Tamarozzi (citation 40 in the updated manuscript). We have edited the limitations subsection of the Discussion section accordingly.

        We sincerely request that the Reviewer offer their experience in identifying important works that may have been overlooked and would contribute to the manuscript's improvement.

“Is it about all hosts? Is it about clinical and diagnosis in people? Does it reach everyone or just Europe? Are you interested in control?”

          As per our stated objectives, our interest is first and foremost in human hosts. When discussing topics of control and prevention (which we consider to include "Real control experiences with different strategies")we have only highlighted and summarized the interventions made, to provide the reader with an extensive overview of current strategies. The presentation of the results of the interventions were never within the declared scope of our paper.

“Then the work advances on certain search and inclusion criteria, not related to the introduction text.

“The inclusion criteria for our research included manuscripts that offered relevant information regarding cystic echinococcosis, such as taxonomic changes, epidemiological data (i.e., human and animal prevalence rates and incidence rates), treatment options, and new research contributing to the control and prevention of the disease (i.e., new biomarkers used for diagnosis, etc.)”.”

          We would like to respectfully disagree with Reviewer’s point on the relationship between search and inclusion criteria and introduction text. Our introduction specifically mentions our focus on epidemiology and the measures taken to properly control the spread of echinococcosis in humans, as well as providing a comprehensive review of recent findings. To achieve this, our inclusion criteria added articles that provide epidemiological data, data in regard to treatment options, and research contributing to disease control and prevention, among others. The exclusion criteria were applied to avoid incorrect agents, as well as to only select original data. For further details, please refer to the Supplementary File S2.

“Real control experiences with different strategies do not appear as eligible.

From this there is a strong and surprising omission of published control experiences, of the use of albendazole in asymptomatic patients and of the systematic use of field ultrasonography. This is associated with a selected bibliography that appears very biased towards certain authors with the omission of others.”

 “Perhaps this would be clearer if the work is directed ONLY to Europe, WHICH would require an overall adjustment of the work.”

          We thank the Reviewer for bringing the matter of including information regarding studies not from the European region. To this end, we have added a Supplementary File (Table S1) to more clearly represent national guidelines in locations with high prevalence rates in South America, Asia and Africa, which we have found to follow the recommendations of the World Health Organization. Furthermore, we strive to include the relevant work of all of our fellow colleagues from around the world in our manuscript, and we regret any omissions we may have made in this regard. To exemplify the methodology and to show our search was not limited to Europe, a view of the database search results was included in the Supplementary File Table S2. Please kindly refer to previous response about articles cited below.

          Concerning albendazole use in asymptomatic patients, we have made changes in the Chemotherapy and “Watch-and-wait” approach portions of the 3.3.2 Treatment subsection, particularly lines 285-288, 297-301 and 316-319 to add more insight into both this matter and the matter of albendazole and praziquantel use in patients in general. To this end, we consider that clinical practice varies depending on multiple factors, which we have mentioned in the full text manuscript, lines: 268-271.

        In regard to the systematic use of field ultrasonography, the data provided in Table 1 has been gathered through ultrasonography population surveys, divided by continent.

        We would once again respectfully wish to ask the Reviewer to share their expertise in identifying essential works that may contribute to the improvement of the manuscript.

          We sincerely thank the Reviewer for their comments and overall evaluation of our manuscript, and we hope that the response herein and the changes made to the manuscript are adequate to the Reviewer’s expectations.

Respectfully,

The Authors

References

  1. Khan, A.; Ahmed, H.; Khan, H.; Saleem, S.; Simsek, S.; Brunetti, E.; Afzal, M.S.; Manciulli, T.; Budke, C.M. Cystic Echinococcosis in Pakistan: A Review of Reported Cases, Diagnosis, and Management. Acta Tropica 2020, 212, 105709, doi:10.1016/j.actatropica.2020.105709.
  2. Colpani, A.; Achilova, O.; D’Alessandro, G.L.; Budke, C.; Mariconti, M.; Muratov, T.; Vola, A.; Mamedov, A.; Giordani, M.T.; Suvonkulov, U.; et al. Ultrasound-Based Prevalence of Cystic Echinococcosis in the Samarkand Region of Uzbekistan: Results from a Field Survey. Am J Trop Med Hyg 2023, 109, 153–158, doi:10.4269/ajtmh.22-0376.
  3. Feng, X.; Qi, X.; Yang, L.; Duan, X.; Fang, B.; Gongsang, Q.; Bartholomot, B.; Vuitton, D.A.; Wen, H.; Craig, P.S. Human Cystic and Alveolar Echinococcosis in the Tibet Autonomous Region (TAR), China. J Helminthol 2015, 89, 671–679, doi:10.1017/S0022149X15000656.
  4. Luo, A.; Wang, H.; Li, J.; Wu, H.; Yang, F.; Fang, P. Epidemic Factors and Control of Hepatic Echinococcosis in Qinghai Province. J. Huazhong Univ. Sci. Technol. [Med. Sci.] 2014, 34, 142–145, doi:10.1007/s11596-014-1246-8.
  5. Paternoster, G.; Boo, G.; Flury, R.; Raimkulov, K.M.; Minbaeva, G.; Usubalieva, J.; Bondarenko, M.; Müllhaupt, B.; Deplazes, P.; Furrer, R.; et al. Association between Environmental and Climatic Risk Factors and the Spatial Distribution of Cystic and Alveolar Echinococcosis in Kyrgyzstan. PLoS Negl Trop Dis 2021, 15, e0009498, doi:10.1371/journal.pntd.0009498.
  6. Acosta-Jamett, G.; Hernández, F.A.; Castro, N.; Tamarozzi, F.; Uchiumi, L.; Salvitti, J.C.; Cueva, M.; Casulli, A. Prevalence Rate and Risk Factors of Human Cystic Echinococcosis: A Cross-Sectional, Community-Based, Abdominal Ultrasound Study in Rural and Urban North-Central Chile. PLoS Negl Trop Dis 2022, 16, e0010280, doi:10.1371/journal.pntd.0010280.
  7. Reyes, M.M.; Taramona, C.P.; Saire-Mendoza, M.; Gavidia, C.M.; Barron, E.; Boufana, B.; Craig, P.S.; Tello, L.; Garcia, H.H.; Santivañez, S.J. Human and Canine Echinococcosis Infection in Informal, Unlicensed Abattoirs in Lima, Peru. PLoS Negl Trop Dis 2012, 6, e1462, doi:10.1371/journal.pntd.0001462.
  8. Li, B.; Quzhen, G.; Xue, C.-Z.; Han, S.; Chen, W.-Q.; Yan, X.-L.; Li, Z.-J.; Quick, M.L.; Huang, Y.; Xiao, N.; et al. Epidemiological Survey of Echinococcosis in Tibet Autonomous Region of China. Infect Dis Poverty 2019, 8, 29, doi:10.1186/s40249-019-0537-5.

Reviewer 4 Report

Comments and Suggestions for Authors

Congratulation to this manuscript based on thousands of data collected throughout the recent literature. Strict threat throughout the whole manuscript, correct conclusions!

Review of the knoweldge of the status presence of the epidemiology and geographic distribution of the species Echinococcus granulosus sensu lato and of the divers genotypes of E. granulosus in the animal and human population of the world. We have a lot of publications on the epidemiology, geographic distribution, clinical symptoms, therapeutic aspects of several countries, but not such a comprehensive presentation on this topic (particularly regrading the tables).

Metaanalyses are as good as the papers used for the study; I think that the authors did their best to interprete the results of their study. I know that it is not easy to include so many data into one study and to present these data clear in a table. I think the authors succeeded with the shown presentation.

Author Response

Esteemed Reviewer,

            We, the Authors, would like to sincerely thank the Reviewer for their appraisal and overall evaluation of our manuscript. We hope that, as mentioned by the Reviewer, our work can provide an adequate presentation of the topic of Cystic Echinococcosis in a manner that is easy to understand and reference.

            We would also like to note that some changes have been made to the original manuscript following the first round of review; we hope that these changes still align with the Reviewer’s expectation in regard to quality, comprehension, and readability. Please refer to the PDF version of the manuscript for the updated line numbers.

            Please note that following the suggestion of the reviewers, Supplementary File: ”Table_S1”, became ”Table_S3”.

Respectfully,

The Authors

Round 2

Reviewer 1 Report

Comments and Suggestions for Authors

I am satisfied with the changes made by the authors and agree with the thoughtful justifications.